# Measurement of ion acceleration and diffusion in a laser-driven magnetized plasma

J. T. Y. Chu [1] ✉, J. W. D. Halliday [1,2], C. Heaton [1], K. Moczulski[3,4], A. Blazevic [5], D. Schumacher[5], M. Metternich[5,6], H. Nazary [5,6], C. D. Arrowsmith [1,3], A. R. Bell[1], K. A. Beyer[7], A. F. A. Bott[1], T. Campbell[1], E. Hansen[4], D. Q. Lamb[8], F. Miniati[9], P. Neumayer[5], C. A. J. Palmer [10], B. Reville [7], A. Reyes[4], S. Sarkar [1], A. Scopatz[4], C. Spindloe[2], C. B. Stuart[1], H. Wen[3], P. Tzeferacos[3,4], R. Bingham [2,11] & G. Gregori [1]

Here we present results from an experiment performed at the GSI Helmholtz Center for Heavy Ion Research. A mono-energetic beam of chromium ions with initial energies of ~ 450 MeV was fired through a magnetized interaction region formed by the collision of two counter-propagating laser-ablated plasma jets. While laser interferometry revealed the absence of strong fluid-scale turbulence, acceleration and diffusion of the beam ions was driven by wave-particle interactions. A possible mechanism is particle acceleration by electrostatic, short scale length kinetic turbulence, such as the lower-hybrid drift instability.

Cosmic rays (CRs) and their extraterrestrial origin were discovered more than a century ago[1], with measurements of the CR energy spectrum up to $10^{20}$ eV[2,3]. The exact mechanism that leads to such high-energy particles remains controversial, as does the nature of the sites where this acceleration occurs. Although many different processes may result in CR acceleration, the present-day understanding is that turbulence plays an essential role in energizing both electrons and ions. The original mechanism of CR acceleration by turbulence or plasma waves was proposed by Fermi[4], where CRs are produced as charged particles gain energy in random scattering events with magnetized clouds. Like turbulence, Fermi acceleration is expected to be ubiquitous in the universe, but laboratory evidence for the process remains elusive.

In addition, inhomogeneous astrophysical environments are susceptible to various instabilities, such as the lower-hybrid drift instability (LHDI)[5] which arises from plasma gradients. These instabilities can accelerate particles, possibly playing an important role in the overall energizing of the thermal background[6–9]. Space probes have detected lower-hybrid waves (LHWs) in the terrestrial magnetotail current sheet and their role in the energy dissipation of electrons[10]. A wave-particle model involving LHWs has been used to explain the presence of accelerated auroral electrons[11] and ions[12], and similarly for the simultaneous acceleration of electrons and ions in solar flares[13,14].

Laboratory astrophysics experiments are an invaluable tool in studying complex phenomena such as these acceleration mechanisms[15–17], used in complement with simulations while addressing some of their limitations. Compared to in situ measurements, where many competing processes can obfuscate the physics that is happening, laboratory plasmas can be designed to study specific mechanisms of interest. This paper presents experimental results[18] from an investigation on ion acceleration and diffusion in a magnetized laser-driven plasma conducted at the GSI Helmholtz Center for Heavy Ion Research. The platform was a modified

[1]Department of Physics, University of Oxford, Oxford OX1 3PU, UK. [2]STFC Rutherford Appleton Laboratory, Didcot OX11 0QX, UK. [3]Laboratory for Laser Energetics, University of Rochester, Rochester 14623, USA. [4]Department of Physics and Astronomy, University of Rochester, Rochester 14627, USA. [5]GSI Helmholtzzentrum für Schwerionenforschung GmbH, Darmstadt 64291, Germany. [6]Technische Universität Darmstadt, Darmstadt 64289, Germany. [7]Max-Planck-Institut für Kernphysik, Postfach 103980, Heidelberg 69029, Germany. [8]Department of Astronomy and Astrophysics, University of Chicago, Chicago 60637, USA. [9]Mach42 Ltd., Oxford OX4 4GP, UK. [10]School of Mathematics and Physics, Queens University Belfast, Belfast BT7 1NN, UK. [11]Department of Physics, University of Strathclyde, Glasgow G4 0NG, UK. ✉e-mail: ting.chu@physics.ox.ac.uk

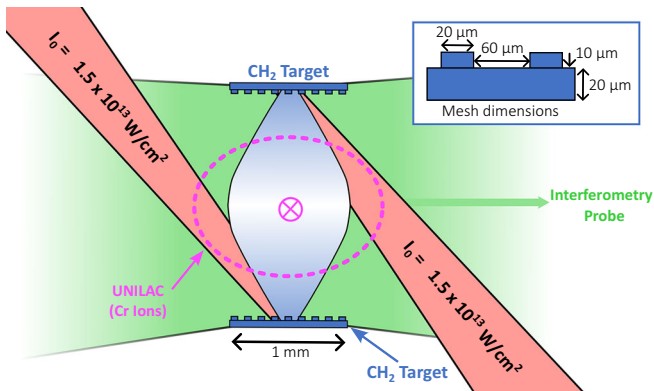

**Fig. 1 | Schematic of the experimental setup fielded at the GSI facility.** The target foils are separated by 1.95 mm, and the dimensions of their mesh structure are shown. The nhelix (nanosecond high-energy laser for heavy ion experiments) laser operating at $\lambda_L = 1053$ nm with a pulse duration of $\tau_L = 10$ ns (approximately linear 3 ns rising and falling edges) is split into two beams of approximately equal energy (25–35J), each illuminating one of the two opposing polypropylene (2:1 ratio of hydrogen atom number to carbon) target foils. The laser spot size is 200 μm on each foil, larger than indicated by the scale in the figure. The UNILAC ion beam direction is into the page, and its approximate size is indicated in magenta. The interferometry laser (green) probes along the axis orthogonal to both the ion beam and target normals, but its size is not shown to scale—the actual field of view spans an area several times larger than the foil separation.

version of that previously fielded to study the turbulent dynamo[19–24], coupled with the heavy ion accelerator at GSI. Plasma conditions were measured and supplemented by results from radiation-magneto-hydrodynamic (MHD) FLASH simulations[25].

## Results

The experimental platform employed for the experiment at the GSI Helmholtz Institute for Heavy Ion research is shown schematically in Fig. 1. Supersonic plasma jets are produced via laser ablation of two opposing target foils, each machined with an embedded grid structure to impose density perturbations. Simultaneously, seed magnetic fields in the azimuthal direction are generated by the Biermann battery mechanism from the cross-product of the density and temperature gradients[26–28]. The two jets collide at the midpoint between the two foils, defined as the target chamber center (TCC), forming an approximately cylindrical magnetized interaction region. The intention of the design was for the initial density perturbations advected by the jets to interpenetrate, introducing shearing motions which would drive turbulent mixing of the plasma. The scale of these instabilities is, however, predicted to be smaller than the experimental resolution of the diagnostics[25]. Additionally, by only driving a single target foil, the case of reduced fluctuations, plasma density, and field strength was accessed for comparison. Interferometry and ion deflectometry were fielded as diagnostics for the experimental plasma conditions.

The monoenergetic ion beam from the UNIversal Linear ACcelerator (UNILAC) was directed through the interaction region, acting as a surrogate for CR particles. The UNILAC supplied a Gaussian pulse train (4 ns width, 28 ns spacing), with only one pulse traversing the interaction region in each shot. Each pulse contains approximately $9 \times 10^5$ ions, a sufficiently low number such that it can be treated as a test beam, with no collective effects expected. The energy and initial charge state of the ion beam were tunable parameters, and we present results obtained using the following: $^{52}Cr^{14+}$ with energies of 8.75 MeV/u; $^{52}Cr^{20+}$ with energies of 8.661 MeV/u.

The ion beam had a diameter of ~1 mm at the TCC, with a divergence of ~10 mrad. A section of the accelerator beamline, located downstream of the laser interaction chamber, was used to direct the ions toward a diamond detector[29,30] situated approximately 12 m

beyond the interaction region. This arrangement enabled the acquisition of ion time-of-flight (ToF) data.

Line-integrated electron densities in the ablated plasma flows and interaction region were measured using laser interferometry, providing a coarse measurement of the plasma density structure. This data also provided an upper bound for the level of fluid turbulence by considering finer density fluctuations below the smaller scale width of the plasma (~0.2 mm). In addition, the density profile of the plasma expansion was used to place a conservative lower bound on the temperature.

The fielded setup consisted of an imaging Mach–Zehnder interferometer and a 355 nm, 500 mJ, 500 ps pulsed probe laser. The data were time-gated by the laser-pulse duration, which is short compared to the system's dynamical timescale ($\tau_{probe} \ll L_p/c_s$ ~10 ns), where $c_s$ is the sound speed and $L_p \simeq 2$ mm is the observed experimental system size, equivalent to the path lengths of the ion beam and interferometry probe through the interaction region. The evolution of the plasma can be tracked with interferometry by changing the probing time in separate shots.

The free electron density, $n_e$, is retrieved from the raw interferograms via the following process. Firstly, the fringes were traced manually to form phase contour maps (Fig. 2A, B). In regions associated with strong density gradients, for example, adjacent to the target foils, individual fringes cannot be resolved—these regions are excluded from the interpolation. Next, the phase contour maps are interpolated to recover a map of the phase shift at each point[31,32]. The phase shift is converted into the line-integrated electron density via the well-known relation:

$$\phi = 2\pi F \approx \frac{\pi}{n_{crit}\lambda_{probe}} \int n_e \mathrm{d}\ell, \tag{1}$$

where $F$ is the number of fringe shifts, $n_{crit}$ is the critical density of the plasma, and $\lambda_{probe}$ is the vacuum wavelength of the probing laser. The result is a 2D map of line-integrated electron density (Fig. 2C). At $t = 10$ ns, a clear interaction region has formed with a width of ~2 mm. Given the approximate azimuthal symmetry of the setup, the path length traversed by the probe beam is also 2 mm, resulting in an average volumetric electron density in the interaction region of $n_e \simeq 3 \times 10^{19}$ cm$^{-3}$. Assuming full ionization with the given plasma conditions, consistent with atomic kinetic simulations, the corresponding ion density $n_i \simeq 1.1 \times 10^{19}$ cm$^{-3}$. Similar analysis for data recorded at $t = 15$ ns gives values of $n_e \simeq 2.5 \times 10^{19}$ cm$^{-3}$ and $n_i \simeq 9 \times 10^{18}$ cm$^{-3}$.

The evolution of the inflow plasma jets was captured by interferometry on shots with only one target foil driven by nhelix, as seen in Fig. 3. The expanding jet can be approximated as isothermal, since the ratio of the temperature equilibration timescale to the experiment duration can be shown to be[33] ~0.05; equivalently, this implies a small Péclet number (Pe $\lesssim$ 0.05). An isothermal expansion has a known density profile[34,35] dependent on the plasma sound speed $c_s$. Since $c_s$ is related to the plasma temperature[36], at a time of $t = 5$ ns, approximately the time the jets collide, the temperature of the inflow jets from this analysis is ~120 eV. Even accounting for collisional heating, we can expect the interaction region plasma to be in the temperature regime of the low hundreds of eV.

Qualitatively, the interferometry data in Fig. 2 show a relatively laminar interaction region, with no evidence of large-scale density fluctuations associated with strong fluid turbulence. To bound any residual fluctuation, we low-pass filter each traced fringe at a cutoff set by the width of the interaction region, take the result as a slowly varying baseline, and subtract it from the raw trace. The residual displacement is then mapped via Eq. (1) to the corresponding fluctuation in line-integrated electron density.

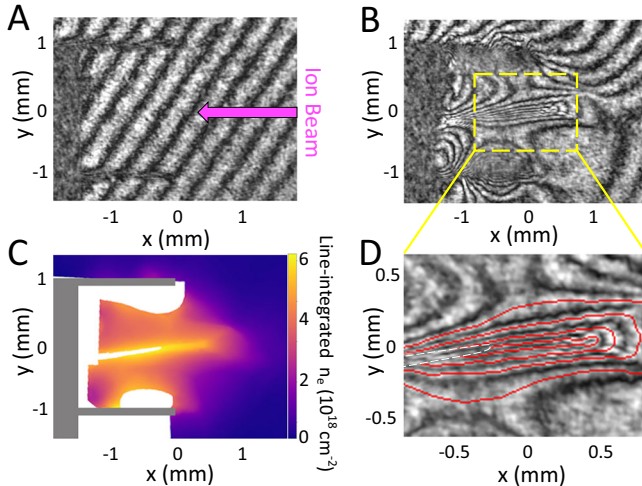

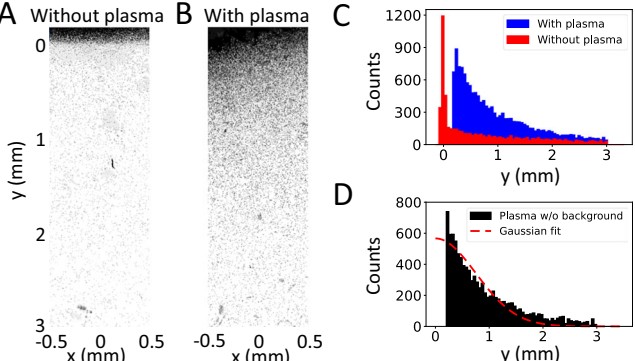

**Fig. 4 | CR-39 data and analysis. A** A background shot with no plasma, cropped to a small region at the periphery of the ion beam. Note the sharp "knife edge" positioned at y = 0 mm, which is exploited in this analysis. **B** A shot where the ion beam traverses the magnetized plasma (double-sided drive) at $t$ = 10 ns, spatially aligned with the background shot. There is a clear increase in pit density away from the knife edge in comparison with the background, as demonstrated in (**C**). **D** The radial distribution of pits moving away from the beam center, with the background subtracted. A Gaussian least-squares fit is shown as the red dashed curve and is used to quantify the magnitude of deflection.

**Fig. 2 | Example of interferometry data at $t$ = 10 ns after the start of the laser drive. A**, **B** Interferograms recorded before and during the shot, respectively; the images have been cropped, and their contrast adjusted for clarity. In addition, a rotation is applied to correct for a small random tilt in the interferometry imaging of ≲5°, such that the target foils are parallel to the *x*-axis of the images. The propagation direction of the ion beam is shown in magenta in (**A**). Processing the interferograms results in (**C**), a map of line-integrated electron density. The projection of the target is filled in with gray; white indicates regions where the density is above the maximum value measurable or where high-density gradients cause self-interference effects that distort the fringe pattern. **D** A close-up of the interaction region bounded by the yellow box in (**B**), overlaid with fringes traced in red.

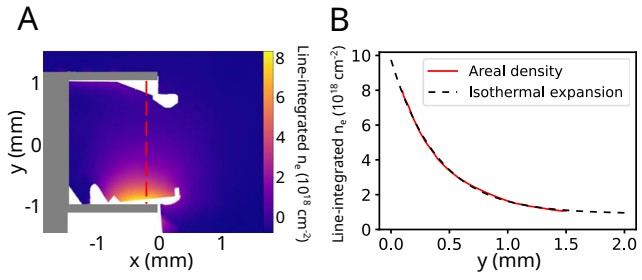

**Fig. 3 | Interferometry data taken at $t$ = 5 ns after the start of laser drive, used to estimate the temperature of each plasma jet prior to collision. A** The line-integrated electron density map, similar to Fig. 2C but only driven from one side. A lineout is taken along the dashed red line in the y direction as displayed in (**B**). The dashed black line shows a least squares fit parameterized by the plasma sound speed, assuming a planar, isothermal expansion.

Next, assuming an isotropic plasma, it can be shown[20] with a random-walk treatment that the line-integrated density fluctuation is related to the volumetric density fluctuation $\delta n_e/n_e$ by:

$$\frac{\langle \int \delta n_e \, d\ell \rangle}{\langle \int n_e \, d\ell \rangle} \sim \frac{\delta n_e}{n_e} \sqrt{\frac{\ell_n}{L_p}}, \tag{2}$$

where $\ell_n$ is the characteristic length scale of the density fluctuations (approximated as the grid spacing $\ell_{grid}$[20]) and $L_p$ is the path length of the interferometry laser through the plasma. The magnitude of $\delta n_e/n_e$ can be used to infer the turbulent velocity $u_{turb}$ using a continuity argument in subsonic, incompressible flow[37]:

$$u_{turb} \simeq c_s \sqrt{\frac{\delta n_e}{n_e}}. \tag{3}$$

Using the values of $c_s$ predicted by MHD simulations[25] for the case of colliding plasma flows, which are more than twice those extracted from the single-sided experimental fits, this results in values of $u_{turb} \lesssim 30$ km s⁻¹ and 30 km s⁻¹ at $t$ = 10 and 15 ns, respectively. These values should be interpreted as an upper bound for $u_{turb}$, as the effect of limited dynamic range, as well as the variability of the manual tracing process, cannot be isolated from actual density fluctuations. This analysis will, in turn, provide upper bounds for the expected magnitude of second-order Fermi acceleration, shown in the later discussion.

Another important parameter relevant to transport properties in the plasma is the magnetic field strength, which was constrained experimentally by fielding an ion deflectometry diagnostic. A nuclear-track detector (CR-39) was placed in the path of the ion beam ~40 cm after the TCC. This was fielded in lieu of the ToF diamond detector, and data were collected both with and without the laser-ablated plasma. For these measurements, the ion beam was gated to the minimum possible length of 10 µs, approximately 350 pulses, to minimize the number of undeflected pulses incident on the plate.

Spatially resolved ion fluence was recorded with this setup. The distribution of pits demarcating the position of ion impacts consists of a saturated central region containing the bulk of the ion beam and a surrounding 'halo' decreasing in pit density with increasing radial distance; Fig. 4A, B shows a section of the data aligned radially outwards from a knife edge positioned ~1 mm from the center of the ion beam, for shots without and with plasma respectively. As the ion beam traverses the plasma, it accumulates a random-walk deflection proportional to the magnetic field strength, which dominates over contributions from the electric fields and Coulomb collisions. The relative contributions from magnetic and electric fields were compared by considering the Lorentz equation of motion and applying resistive MHD; the ratio of deflection force from electric and magnetic fields $\propto u/v_c \lesssim 10^{-2}$, where $u$ is the characteristic fluid velocity ($u \lesssim 200$ km s⁻¹) bounded by estimating the plasma jet displacement between subsequent interferograms, and $v_c$ the initial ion beam velocity.

Since the beam ion gyroradius $r_c$ ~ 0.3 m is much larger than the magnetic coherence length $\ell_B$ ~ 10⁻⁴ m, the ions are unmagnetized, and their deflection accumulates in a series of small-angle scattering events as a random walk process; this manifests as a Gaussian deflection as a consequence of the central limit theorem. The average angle of

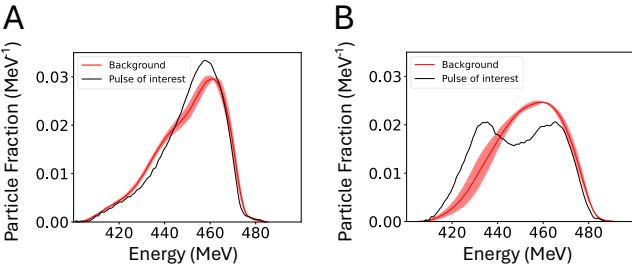

**Fig. 5 | Examples of ion energy spectra extracted from time-of-flight data, probing the interaction region at $t$ = 10 ns.** The background is obtained by averaging the 6 pulses before the pulse of interest (PoI), with the standard deviation shaded in red. Normalization is performed such that the area below the background and PoI profiles equates to 1. **A** Spectra with a small change in shape between the PoI and background pulses. **B** A much larger change, with the PoI exhibiting two distinct peaks. There is noticeable variance between the averaged background pulses in (**A**) and (**B**), which precludes direct comparison of spectra between shots.

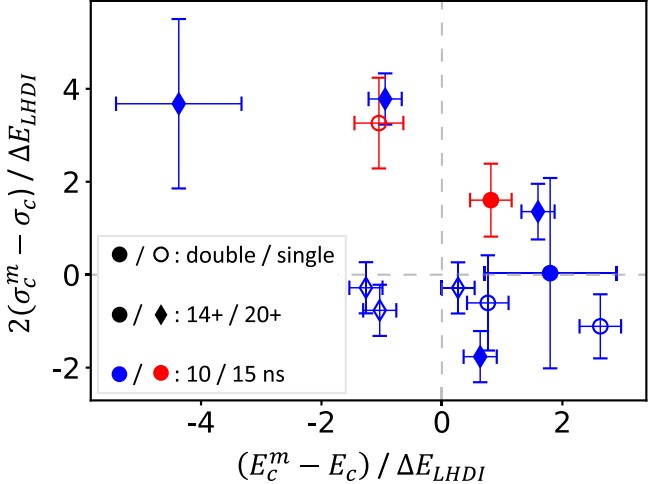

**Fig. 6 | Scatter plot showing energy shift ($E_c^m - E_c$) against width change 2($\sigma_c^m - \sigma_c$).** Filled and hollow points indicate double and single-sided drive, respectively; circular points represent shots with initial ion charges of 14+, whereas diamonds are 20+; blue and red indicate $t$ = 10 and 15 ns, respectively. Error bars are calculated from the standard deviation of the averaged background (shaded red region in Fig. 5). Axes are normalized for the two initial charge states to predicted $\Delta E_{LHDI}$ values at $t$ = 10 ns for double-sided drive. This can be calculated as $\Delta E_{LHDI}$ [MeV] = $1.09\,\overline{Z_c}\,\overline{A_c}[u]^{-1}\overline{B}_{rms}$ [kG] $\overline{c_s}$ [cm s$^{-1}$] $\overline{L_p}$[mm]$^{\frac{1}{2}}\overline{E_c}$[MeV]$^{\frac{1}{4}}\overline{n_e}$[cm$^{-3}$]$^{-\frac{1}{4}}$, where overlined variables are scaled to convenient units as follows: $Z_c$ = 20.3, $A_c$ = 52 u, $B_{rms}$ = 230 kG, $c_s$ = 1.1 × 10$^7$ cm s$^{-1}$, $L_p$ = 2 mm, $E_c$ = 450 MeV, and $n_e$ = 3 × 10$^{19}$ cm$^{-3}$.

deflection is[38]

$$\langle\delta\theta^2\rangle \sim N\left(\frac{\ell_B}{r_c}\right)^2, \qquad (4)$$

where $N$ is the total number of scattering events and can be approximated as $N \sim L_p/\ell_B$. The RMS deflection angle is thus[39]

$$\delta\theta_B \simeq Z_c eB\sqrt{\frac{L_p\ell_B}{2m_c E_c}}, \qquad (5)$$

where $Z_c$, $m_c$, and $E_c$ are the charge state, mass and energy of the beam ions. The magnetic coherence length $\ell_B$ is the characteristic length over which the stochastic magnetic fields maintain coherence and is essentially a measure of the size of the stochastic field structures. An

upper estimate for $\ell_B$ is found by taking the grid spacing (60 μm) on the target foils and accounting for the approximate doubling in spatial extent of the plasma: $\ell_B \lesssim 0.12$ mm. This is on the order of the outer scale of turbulence, which in previous experiments (on similar setups) has been larger than $\ell_B$[20]. Using the experimental value of $\delta\theta_B \simeq 0.1°$ obtained from the width of the distribution in Fig. 4D, this gives a minimum field strength of $B_{min}$ = 40 kG. An upper bound for the field strength can be defined using a simple geometric argument: assuming all the counts in the distribution of Fig. 4D come from ions at the center of the beam, we obtain $\delta\theta_{B,max} \simeq 0.6°$, which is found by summing the initial ion beam divergence and the measured $\delta\theta_B$. This corresponds to a field strength of $B_{max}$ = 230 kG. Thus, we estimate $40 \lesssim B_{rms} \lesssim 230$ kG, which has implications for later discussion on acceleration from fluid and short-scale electrostatic turbulence.

The ToF diamond detector fielded allows for an energy measurement of each pulse within the ion beam. Figure 5A, B presents the resulting energy spectra, comparing the pulse of interest (PoI), that is, the ion pulse that coincides with the plasma, with a background averaged over pulses acquired prior to plasma formation. There is significant shot-to-shot variation even for shots with similar plasma conditions probed at the same time, reflecting the inherently stochastic nature of the beam-plasma interaction. To quantitatively compare shots across the campaign, two main metrics were used: the shift in mean energy of the pulse $E_c^m - E_c$, calculated by integrating the energy profile to find its center-of-mass; and the change in the energy spread of the pulse, defined as twice the standard deviation $\sigma_c^m$ of the pulse around its mean energy.

Figure 6 shows the distribution of the measured acceleration and diffusion from this experiment, revealing a clear distinction between single- and double-sided drive configurations. Single-sided shots cluster near the origin, exhibiting minimal energy shift and broadening, whereas double-sided shots trace a broader curve with significantly larger magnitudes in both metrics. Nevertheless, outliers are present in both cases, reflecting the inherent stochasticity of the beam-plasma interaction. Notably, the double-sided data appear to be divided into two regimes: one characterized by net acceleration with minimal broadening, and another by increased width accompanied by a reduction in mean energy.

In addition, charge transfer processes between the beam and plasma ions introduce a non-trivial correction to $Z_c$ as the ion beam traverses the plasma. To account for this effect, a semi-empirical model[40] based on the Gus'kov model[41] was employed to estimate the mean non-equilibrium charge state of the beam ions. This upshifted value is accounted for in the $Z_c$ estimate (see Table 1).

One possible explanation for deceleration is the charged-particle stopping power of the plasma—the magnitude of this effect can be calculated following a generalized Fokker–Planck equation[42,43] which accounts for both small and large-angle binary collisions:

$$\frac{dE}{dx} \simeq -\frac{(Z_c)^2}{v_c^2}\omega_p^2 ln\Lambda_b, \qquad (6)$$

where $v_c$ is the initial ion beam velocity, $\omega_p$ is the plasma frequency, and $ln\Lambda_b$ is the Coulomb logarithm. Using the experimentally constrained plasma conditions in Table 1, the electron-ion stopping dominates, with values of −0.37 and −0.58 MeV for $Z_c$ = 16.1 and $Z_c$ = 20.3, respectively, assuming a path length of 2 mm. This magnitude of deceleration cannot account for the observed shifts in mean energy. Similarly, the expected parallel diffusion associated with Coulomb collisions is found to be negligible (≪1 eV) under the experimental conditions, due to the high initial velocity of the ion beam.

The results can be compared with theoretical predictions, considering two candidate mechanisms informed by previous theoretical

**Table 1 | Table summarizing important plasma parameters at different times for both drive configurations**

| Drive | Single | | | Double | |
|---|---|---|---|---|---|
| $t$ (ns) | 5 | 10 | 15 | 10 | 15 |
| $n_e$ (cm$^{-3}$) | $9 \times 10^{18}$ | $9 \times 10^{18}$ | $8 \times 10^{18}$ | $3 \times 10^{19}$ | $2.5 \times 10^{19}$ |
| $n_i$ (cm$^{-3}$) | $3 \times 10^{18}$ | $3 \times 10^{18}$ | $3 \times 10^{18}$ | $1.1 \times 10^{19}$ | $9 \times 10^{18}$ |
| $T$ (eV) | 120 | 50 | 10 | – | – |
| $B$ (kG) | – | – | – | 40–230 | – |
| $u_{turb}$ (cm s$^{-1}$) | $\lesssim 1.8 \times 10^6$ | $\lesssim 8 \times 10^5$ | $\lesssim 4 \times 10^5$ | $\lesssim 3 \times 10^6$ | $\lesssim 2.6 \times 10^6$ |
| $c_s$ (cm s$^{-1}$) | $8 \times 10^6$ | $5 \times 10^6$ | $2 \times 10^6$ | – | – |
| $n_e^{sim}$ (cm$^{-3}$) | $2.2 \times 10^{19}$ | – | – | $2.4 \times 10^{19}$ | $2.15 \times 10^{19}$ |
| $n_i^{sim}$ (cm$^{-3}$) | $6.7 \times 10^{18}$ | – | – | $9.3 \times 10^{18}$ | $7.1 \times 10^{18}$ |
| $T_{sim}$ (eV) | 412 | – | – | 106 | 80 |
| $B_{sim}$ (kG) | 38 | — | — | 77 | 40 |
| $c_s^{sim}$ (cm s$^{-1}$) | $1.7 \times 10^7$ | — | — | $1.1 \times 10^7$ | $8.4 \times 10^6$ |
| $Z_c$ | $Z_0 = 14$ | 14.7 | 14.6 | 16.1 | 15.8 |
| | $Z_0 = 20$ | 20.1 | 20.1 | 20.3 | 20.3 |
| $\Delta E_{Fermi}$ (keV) | $Z_0 = 14$ | 0.2 | 0.1 | 10.3 | 2.5 |
| | $Z_0 = 20$ | 0.3 | 0.2 | 13.1 | 3.1 |
| $\Delta E_{LHDI}$ (keV) | $Z_0 = 14$ | 140 | 60 | 870 | 300 |
| | $Z_0 = 20$ | 170 | 80 | $1.09 \times 10^3$ | 390 |

Experimentally measured or inferred values at $t$ = 5, 10 and 15 ns are compared to results from 3D FLASH magnetohydrodynamics simulations[1]. These values are used to calculate the corrected charge state $Z_c$ as well as the expected $\Delta E$ according to Eq. (7) and (10), for initial charge states $Z_0$ = 14 and 20 at $t$ = 10 and 15 ns.
[1]FLASH simulation data was retrieved with the following method. Values are averaged over a 0.95 mm radius cylinder centered between the targets, which extends the extent of the domain. A temperature threshold is used to exclude solid target material. Values are averaged over two runs, with 25J and 35J of laser energy on each target foil, respectively.

work[25]. The first is second-order Fermi acceleration[44], originally proposed to account for the origin of high-energy cosmic rays by interactions between particles and mirror fields in the interstellar medium[4]. This process can be generalized to a diffusive interaction between charged particles and magnetized plasma turbulence. For this experiment, as the ion beam traverses the interaction region, the ions undergo many deflections by magnetic fluctuations, which result in a total energy change of

$$\Delta E_{Fermi} \sim \frac{Z_c e u_{rms} B_{rms} \sqrt{L_p \ell_B}}{c}, \quad (7)$$

where $c$ is the speed of light and $u_{rms}$ and $B_{rms}$ are the RMS velocity and magnetic field, respectively. Taking $B_{rms} \simeq B_{max}$ (bounded by deflectometry) and $u_{rms} \simeq u_{turb}$ (bounded by interferometry), the maximum $\Delta E_{Fermi}$ associated with second-order Fermi acceleration (see Table 1) is much smaller than the initial energy spread of the ion beam ($\lesssim 0.1\%$ 0.4 MeV) and the energy resolution of the diamond detector. As $T$ and hence $u_{turb}$ are only bounded from below by interferometry, one might consider the effect of higher temperature on the magnitude of $\Delta E_{Fermi}$. A conservative back-of-the-envelope scaling ($u_{turb} \propto T^{1/2}$) shows that one would need temperatures of order $\gtrsim 750$ keV before the Fermi term could become competitive—clearly not physical for this platform and inconsistent with the observed density and expansion dynamics. Consequently, while this mechanism may contribute, it cannot be responsible for the changes measured—a conclusion that is robust to substantially higher local temperatures.

The second possibility is a wave-particle interaction, such as the lower-hybrid drift instability. LHDI is a kinetic instability driven by strong density and magnetic field gradients[5], which can excite lower-hybrid waves (LHWs) that in turn induce an electric field for ion

acceleration[6]. To demonstrate that the experimental data are not inconsistent with the formation of the instability, we infer an upper bound for the LHDI growth rate using experimental values. The maximum linear growth rate of this instability $\gamma_{max}$ is given by

$$\gamma_{max} = \frac{\sqrt{2\pi}}{8} (\epsilon_n r_i)^2 \omega_{LH}, \quad (8)$$

where $r_i \sim 0.2$ mm is the gyroradius of the plasma ions, $\omega_{LH}$ is the lower-hybrid frequency, and $\epsilon_n$ is the inverse gradient scale length, which can be approximated as

$$\epsilon_n = \frac{1}{n_e} \frac{\partial n_e}{\partial x} \sim \frac{1}{n_e} \frac{\delta n_e}{\ell_{grid}}. \quad (9)$$

Using the magnitude of density fluctuations bounded by interferometry, $\gamma_{max} \simeq 6.2 \times 10^8$ s$^{-1}$ at $t$ = 10 ns, corresponding to an e-folding time of 1.6 ns. Additionally, the gradient scale length $L_n = 1/\epsilon_n \sim 0.6$ mm satisfies the steepness condition $L_n/r_i < (m_i/m_e)^{1/4}$[45] required to excite LHDI. Note that this analysis only includes density gradients associated with resolvable fluctuations, limited by the dynamic range of the interferometry diagnostic. Gradients on a smaller scale, closer to $r_i$, would instead excite kinetic instabilities, with higher growth rates on the order of $\omega_{LH} \sim 1.24 \times 10^{10}$ rad s$^{-1}$[45–47].

In the context of this experiment, LHWs are assumed to interact linearly and non-resonantly with the ion beam in a weakly turbulent plasma[48]. This can be treated as a diffusion problem, with an associated energy change of

$$\Delta E_{LHDI} \simeq \kappa Z_c A_c^{-1} B_{rms} c_s (v_c L_p)^{\frac{1}{2}} n_e^{-\frac{1}{4}}, \quad (10)$$

where $\kappa \sim 4 \times 10^{-14}$ C m$^{-\frac{3}{4}}$s$^{-\frac{1}{2}}$ is a constant of proportionality that captures the level of lower-hybrid turbulence (see ref. 25 for details) and $A_c = 52$ u the atomic mass of the beam ions. The theoretically predicted values (Table 1) for LHDI are sufficiently high for measurement and are expected to dominate over the effect of second-order Fermi acceleration.

An alternative avenue for the generation of electrostatic waves near the lower-hybrid frequency is via beam-driven instabilities, such as the modified two-stream instability (MTSI)[49]. To demonstrate that the ion beam acts as a test beam and does not excite plasma instabilities, we consider the MTSI growth rate: the ion beam density of ~$10^7$ cm$^{-3}$ results in growth rates of the order $10^5$–$10^6$ s$^{-1}$. This is much smaller than the growth rate for the lower-hybrid drift instability, and therefore, we discount the possibility of turbulence being driven by the ion beam.

The observation of bulk energization cannot be explained by the broadening process outlined above alone, and suggests that coherent fields which facilitate particle acceleration are generated in the plasma. A tantalizing possibility consistent with our experiment is the role of short-scale electrostatic turbulence (below the interferometry diagnostic's resolution) in developing coherent structures within the plasma. In literature, both Langmuir wave[50] and LHW[51,52] turbulence have been shown to drive the evolution of these wave structures. On the other hand, we can also consider electromagnetic turbulence, which can be driven by plasma instabilities such as the ion Weibel instability and ion filamentation instability (IFI)[53]. To investigate this possibility, 1D particle-in-cell (PIC) simulations using the OSIRIS PIC code were performed[25], showing that the counter-streaming phase is short-lived (3 ns) compared to the e-folding times of the IFI and Weibel instabilities (~10 ns). It is therefore unlikely that these instabilities contribute to field amplification and ion acceleration.

In summary, we present observations of ion acceleration and diffusion in a laser-driven magnetized plasma. Despite the absence of large-scale fluid turbulence as diagnosed by interferometry, evidence for ion

acceleration and energy diffusion was observed with a time-of-flight detector, suggesting that a wave-particle interaction was the underlying acceleration mechanism. Analysis showed that the lower-hybrid drift instability was consistent with the measured acceleration and diffusion, although other wave-particle interactions could also be responsible.

## Methods

### Retrieval of free electron density from interferometry using MAGIC

The free electron density is extracted from the interferometry data with the help of MAGIC, an analysis software (available at https://github.com/jdranczewski/Magic2).

First, in an image-editing software, the interferograms (one captured during the plasma interaction and one reference image taken before the interaction) are cropped to a region containing both the interaction plasma as well as a region containing no plasma—this defines a point with zero fringe shift that allows for the absolute density to be calculated. Next, the position of the fringe intensity minima and/or maxima is manually traced to produce the specially formatted image files required by the MAGIC software as inputs. The fringes in the interaction region can be complicated, and tracing them requires a certain degree of human interpretation and experience. The fringes are automatically scanned by the software, and then we label the fringes in order of increasing or decreasing phase. The software performs a 2D linear triangular interpolation between the fringes to produce phase maps; subtracting the reference from the data produces a map of phase shift, which is converted to line-integrated electron density via Ēq. (1). This requires knowledge of the real spatial size corresponding to one pixel width, which is estimated using the known distance between the two target foils.

### CR-39 processing

The CR-39 plates are etched with NaOH solution, forming pits that reveal the location of molecular damage caused by ion impacts. The plates are then scanned with an optical microscope to produce high magnification images.

The two images (one with and one without the ion beam traversing the plasma) are spatially aligned and cropped to a region containing the "knife edge". This region contains on the order of $10^4$ pits in each of the images. A coordinate system is established where the $y$ direction aligns radially away from the center of the beam, and the pits are manually tagged in an image-editing software, with the coordinates of each pit recorded. As the pit density increases towards the center of the beam, pits can start overlapping, and a degree of interpretation is required.

### Time-of-flight data processing

The ToF data has to be preprocessed before any analysis can be performed. Firstly, there is an X-ray "flash" created by the laser drive of the target foils, which is recorded by the diamond detector. To remove this effect, the response of the X-ray flash on the diamond detector was isolated by performing background shots with only laser drive but no ion beam and subtracted from the signal. In addition, there is a slow-varying amplitude modulation of the signal associated with source instability on the accelerator side. To correct for this, two cubic splines are fitted to the signal envelope and used to normalize the signal between 0 and 1.

Next, the normalized signal is transformed into energy profiles for each pulse by performing a change of variables from $t$ to $E$ using the relativistic relation:

$$E(t) = m_c c^2 \left( \sqrt{\frac{1}{1 - \left(\frac{d}{ct}\right)^2}} - 1 \right), \tag{11}$$

where $d$ is the distance from the plasma to the diamond detector. To reverse the distortion introduced by this transformation, the signal is multiplied by the 1D Jacobian $\frac{dt}{dE}$.

## Data availability

The data generated in this study have been deposited in the figshare database under accession code https://doi.org/10.6084/m9.figshare.30024130.

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

## Acknowledgements

This project has received funding from the European Union's Horizon 2020 research and innovation program under grant agreement no. 871124 Laserlab Europe, the UK Research and Innovation (UKRI) Frontier Research Guarantee under Grant No. EP/Y035038/1 (JET-LAB), and by the Central Laser Facility Vulcan dark period community support program 24-1. The results presented here are based on the experiment P-22-00089, which was performed at the target station Z6 at the GSI Helmholtzzentrum fuer Schwerionenforschung, Darmstadt (Germany) in the frame of FAIR Phase-0.

## Author contributions

This project was conceived by G.G., P.T., and A.F.A.B. The experiment was designed by J.T.Y.C., J.W.D.H., G.G., P.T., and A.F.A.B. and carried out by J.T.Y.C., J.W.D.H., C.H., K.M., A.B., D.S., M.M., and H.N. The data analysis was carried out by J.T.Y.C. with support from J.W.D.H., R.B., G.G., C.H., and A.F.A.B. The manuscript was written by J.T.Y.C., with input from J.W.D.H., R.B., G.G., C.A.J.P., B.R., K.A.B., and A.F.A.B. Numerical simulations were performed by K.M. and P.T. Further experimental and theoretical support was provided by C.D.A., A.R.B., T.C., E.H., D.Q.L., F.M., P.N., A.R., S.S., A.S., C.S., C.B.S., and H.W.

## Competing interests

The authors declare no competing interests.
