## [Transparent Peer Review file · Nature Communications]

Measurement of ion acceleration and diffusion in a laser-driven magnetized plasma

Corresponding Author: Mr Joshua Ting Yu Chu

Version 0:

Reviewer comments:

Reviewer #1

(Remarks to the Author)

This manuscript presents an important experimental result which shows that, even in the absence of strong fluid-like turbulence, ion acceleration and diffusion can be driven in a magnetized plasma by wave particle interactions.

This result is relevant to the general field of plasma turbulence, both in the laboratory and in space, in particular in the context of cosmic ray acceleration.

The manuscript presents a thorough quantitative analysis of the different physical mechanisms that might explain the experimental findings. In the last part of the manuscript a possible mechanism is proposed based on the role of short scale electrostatic turbulence. However this hypothesis cannot be substantiated because such a turbulence would occur on scale-lengths below the interferometric resolution of the experiment.

I found the manuscript clear and well reasoned and I would suggest to accept it for publication as is even if additional investigations will be needed in order to verify the role of short scale electrostatic turbulence and, in particular, of the lower-hybrid drift instability.

(Remarks on code availability)

Reviewer #2

(Remarks to the Author)

Comments for the Authors

General comments.

The authors of the manuscript "Measurement of ion acceleration and diffusion in a laser-driven magnetized plasma" did an original study and the results very interesting.

The authors show that the used ions beam energization and diffusion could occur experimentally in laser-driven colliding plasma due mainly to the lower hybrid wave mechanism. The results presented are quite new.

This work is well supported experimentally and well based on modeling and simulations.

It is a well-written and -structured manuscript with clear explanations.

However, I have some questions to clarify some parts.

In experimental set-up part : The authors present the ion beam characteristics of ion beam from UNILAC, could you give the number of ions in each bunch of 4 ns ?

In Fig. 1 caption : I propose to replace 200 um by 200 μm

Could state about the choose the target grid spacing ?

In results part : Could you give reference to the Figure 2 before the equation [1] ?
On the Fig.2 : Could you give the direction of the ion beam ?

In the Caption of Fig. 2 : in the sentence : “ the images have been cropped, rotated, and their contrast”, why the image is rotated ?

Could the give reference to Fig. 3 perhaps in the following paragraph :“The evolution of the inflow plasma jets was captured by interferometry on shots with only one target foil driven by nhelix. The ...”

In sentence “ the ratio of deflection force from electric and magnetic fields $\propto u/v_c \lesssim 10^{-2}$, where u is the characteristic fluid velocity and v_c the initial ion beam velocity.”
How did you estimate this ratio (u/v_c)?

In sentence “Since the beam ion gyroradius $r_c \sim 0.3$ m is much larger than the magnetic coherence length $l_B \sim 10^{-4}$ m,” ;
The authors talk about r_c but in the Eq.[4] you talk about r_g ?
What is the magnetic coherence length l_B ?

In the sentence : “An upper estimate for the magnetic coherence length is found by taking the grid spacing (60 μ m) “ , I suggest to replace 60 μ m by 60 μ m.

After the expression [6] I suggest to recall that v_c is the ion velocity.

In the Table I : How did you measure or estimate the magnetic field values: 40-230 kG for 10 ns (double drive) ?

(Remarks on code availability)

Reviewer #3

(Remarks to the Author)

The manuscript reports intriguing experimental results that integrate laser-driven colliding plasmas with large-scale heavy-ion accelerators. The laser-driven colliding plasmas are employed to simulate electromagnetic turbulence, a phenomenon ubiquitous in the universe. Meanwhile, the heavy-ion beams serve as proxies for cosmic rays. Plasma density is measured using laser interferometry, and the density profile is reconstructed through isothermal expansion modeling. The key finding is the significant acceleration and broadening of ion beams during their transport through plasmas, which the authors hypothesize may be linked to the origin of high-energy cosmic rays.

While the topic is compelling and the manuscript is well-written, I do not believe it is suitable for Nature Communications in its current form. Below, I outline my concerns.

1) The diagnostic approach for plasma conditions is inadequate, as it relies solely on laser interferometry, which is sensitive only to plasma density. While the plasma expansion can be modeled using isothermal assumptions, the temperature evolution during colliding processes is both complex and critical for instability analysis. The absence of reliable temperature diagnostics constitutes a major weakness of this manuscript.

2) In Fig. 5B, the claimed ion beam acceleration is not clearly discernible.

3) In P. Liu et al., Phys. Rev. Letters 132, 155103 (2024), full kinetic simulations were performed, revealing distinct instability mechanisms and turbulence. The authors should compare their results with this study, particularly since the FLASH code likely cannot capture the colliding plasma dynamics accurately—mutual penetration and filamentation turbulence are inherently kinetic processes.

4) In Ren et al., Nature Communications 11, 5157 (2020), a similar experimental setup demonstrated enhanced stopping power of intense proton beams due to collective beam effects. The authors should also compare their findings with this work, emphasizing the role of heavy-ion beam flux as a key parameter.

(Remarks on code availability)

Version 1:

Reviewer comments:

Reviewer #2

(Remarks to the Author)

For my side the authors have answer well at all my comments and questions. Then I suggest to accept the manuscript for

publication.

(Remarks on code availability)

Reviewer #3

(Remarks to the Author)

The authors had well addressed my comments. I believe now the paper can be accepted.

(Remarks on code availability)

Response to Referees

NCOMMS-25-69449: “Measurement of ion acceleration and diffusion in a laser-driven magnetized plasma”

J. T. Y. Chu et al.

November 24, 2025

We thank all three reviewers for a careful reading of our manuscript and for their constructive comments. Reviewer #1 is supportive and recommends acceptance as is, noting the relevance of our findings to plasma turbulence and cosmic-ray acceleration. Reviewer #2 is positive about the novelty and experimental support, suggesting clarifications and minor edits (figure captions/references, notation consistency, units, ion-bunch details, rationale for grid spacing, and the derivation of certain estimates). Reviewer #3 finds the topic compelling but raises four concerns: the adequacy of temperature constraints, the visibility of acceleration in Fig. 5b, and comparisons to both fully kinetic simulations (Liu et al., PRL 2024) and collective stopping in a related geometry (Ren et al., Nat. Commun. 2020). We address each point carefully below in a point-by-point response and have made targeted revisions to improve clarity and completeness.

All changes from the previous version of the manuscript have been highlighted in red in the current version. Figures 2 and 6 have also been modified slightly: the ion beam direction was added to Fig. 2A, and a mistake in the y-axis label in Fig. 6 was rectified.

Reviewer 1

Reviewer 1 does not suggest any substantial changes to the manuscript – we thank them again for their careful review.

Reviewer 2

Response to minor comments 1, 3, 5, 8, 9:

Comments 1, 3, 5, 8, 9 by Reviewer 2 deal with minor edits, including typographic improvements, clarification of variables, and adding references to figures within the text. We thank the reviewer for making these suggestions and agree that they improve the readability of the paper – all suggestions have been incorporated into the manuscript.

Response to more substantial comments:

Comment 2:

Could state about the choose the target grid spacing ?

Response:

As mentioned in the manuscript, the our intention was to drive fluid turbulence as the ablation plasma from the two foils collide. The grid spacing was informed from numerical simulation (Moczulski et al., Phys. Plasmas 31, 122105 (2024)). However, the experiment showed that the level of fluid turbulence achieved was insufficient to explain the observed acceleration and energy broadening of the ion beam. This motivated us to look for alternative explanations, such as those related to short-scale electrostatic turbulence.

Comment 4:

In the Caption of Fig. 2 : in the sentence : “ the images have been cropped, rotated, and their contrast”, why the image is rotated ?

Response:

There is a small random tilt in the interferometry imaging of $\lesssim 5^\circ$. A rotation is applied to correct for this tilt, such that the target foils are parallel to the x-axis of the images. We have added this information in the caption of Fig. 2.

Comment 6:

In sentence “ the ratio of deflection force from electric and magnetic fields $\propto u/v_c \lesssim 10^{-2}$, where u is the characteristic fluid velocity and v_c the initial ion beam velocity.” How did you estimate this ratio (u/v_c)?

Response:

An upper limit for the fluid velocity $u \lesssim 200$ km/s is found by estimating the supersonic plasma jet velocity from interferometry data. This is done by looking at how much the plasma jet moves between subsequent timesteps. $v_c = 4 \times 10^4$ km/s is a known quantity given to us by the UNILAC operators. This has now been clarified in the manuscript.

Comment 7:

In sentence “Since the beam ion gyroradius $r_c \sim 0.3$ m is much larger than the magnetic coherence length $\ell_B \sim 10^{-4}$ m,” ; The authors talk about r_c but in the Eq.[4] you talk about r_g ? What is the magnetic coherence length ℓ_B ?

Response:

We appreciate the reviewer catching this inconsistency, eq. (4) has been amended to use r_c as well. ℓ_B is the characteristic length over which the stochastic magnetic fields maintain coherency and is essentially a measure of the size of the stochastic field structures. We can estimate an upper bound for this as the size of the initial density perturbations, i.e. the grid spacing on the target foils. We have also added this definition in the manuscript for clarity.

Comment 10:

In the Table I : How did you measure or estimate the magnetic field values: 40-230 kG for 10 ns (double drive) ?

Response:

These values were measured using the CR-39 detector. The method is outlined in the text after eq. (5).

Reviewer 3**Comment 1:**

The diagnostic approach for plasma conditions is inadequate, as it relies solely on laser interferometry, which is sensitive only to plasma density. While the plasma expansion can be modeled using isothermal assumptions, the temperature evolution during colliding processes is both complex and

critical for instability analysis. The absence of reliable temperature diagnostics constitutes a major weakness of this manuscript.

Response:

We agree that a direct measurement of temperature is important, but believe that it is sufficiently constrained by our current analysis. The interferometry-based isothermal expansion fit provides a *lower bound* on the inflow temperature of ~ 120 eV at the time the jets meet, and independent FLASH simulations place the interaction region in the few-hundred-eV regime, consistent with and above that bound. Thus, the temperature is bracketed by experiment and modeling rather than left unconstrained.

The only place where a higher temperature could impact our conclusions is through the upper bound on the turbulent velocity used to limit second-order Fermi acceleration. In the manuscript we estimate

$$u_{\text{turb}} \lesssim c_s \sqrt{\delta n_e / n_e}, \quad c_s \propto T^{1/2},$$

so u_{turb} – and hence $\Delta E_{\text{Fermi}} \propto u_{\text{turb}} B_{\text{rms}}$ – depends only *weakly* on T (square-root scaling). Using the measured $\delta n_e / n_e$ and the magnetic-deflectometry bounds on B_{rms} , our predicted ΔE_{Fermi} remains far below the observed energisation across the entire plausible temperature range. A conservative back-of-the-envelope scaling shows that one would need temperatures of order $\gtrsim 750$ keV before the Fermi term could become competitive – clearly not physical for this platform and inconsistent with the observed density and expansion dynamics.

By contrast, the wave-particle pathway we discuss (e.g. LHDI-mediated energisation) is *not* weakened by higher T ; the relevant scalings are either insensitive to T or become more favorable as c_s increases, so the hierarchy between mechanisms is preserved. To make this explicit, we have added a shortened version of the above discussion into the manuscript text: (i) stating in the results section that interferometry provides “a conservative lower bound on temperature” and (ii) that the conclusions are “robust to substantially higher local temperatures”.

Comment 2:

In Fig. 5B, the claimed ion beam acceleration is not clearly discernible.

Response:

While the acceleration might not be visually discernible in Fig. 5B, we can quantify the level of acceleration and energy broadening using the two metrics outlined in the paper: “the shift in mean energy of the pulse $E_c^m - E_c$, calculated by integrating the energy profile to find its center-of-mass; and the change in the energy spread of the pulse, defined as twice the standard deviation σ_c^m of the pulse around its mean energy.” These metrics shows that there is a statistically significant mean shift in energy of $E_c^m - E_c = -4.76$ MeV and a broadening of $2(\sigma_c^m - \sigma_c) = 4.01$ MeV.

Comment 3:

In P. Liu et al., Phys. Rev. Letters 132, 155103 (2024), full kinetic simulations were performed, revealing distinct instability mechanisms and turbulence. The authors should compare their results with this study, particularly since the FLASH code likely cannot capture the colliding plasma dynamics accurately—mutual penetration and filamentation turbulence are inherently kinetic processes.

Response:

We appreciate the reviewer for bringing this paper to our attention and agree that the possibility of electromagnetic turbulence playing a role warrants careful consideration given the similarities between the setup in Liu et al. 2024 and ours. As the reviewer points out, FLASH is a MHD code and *cannot* capture the kinetic effects that lead to the development of ion filamentation and Weibel instabilities. Thus, we refer to 1D OSIRIS particle-in-cell simulations performed using initial conditions relevant to our experimental setup (see K. Moczulski et al., Phys. Plasmas 31 (12), 122105 (2024)). These

simulations show that the counter-streaming phase of the interaction between the two plasma jets is short-lived: it lasts for 3 ns after the two jets start to merge. Since these 1D simulations result in a longer counter-streaming time than the 3D case, this gives an *upper-bound* for the duration of the counter-streaming phase.

We can then compare this counter-streaming duration to the expected growth times of the relevant instabilities. For the ion Weibel instability, K. Moczulski et al. (2024) calculates an e-folding time of 6.7 ns. As this is longer than the duration of the counter-streaming phase, it is *unlikely* for this instability to contribute to field amplification and driving turbulence. Similarly, we can estimate the growth time of the ion filamentation instability (IFI). In P. Liu et al. (2024), they calculate a growing-up timescale for the IFI to be ~ 100 ps, using a similar temperature (100 eV) and electron density ($4 \times 10^{19} \text{ cm}^{-3}$). The main difference between their work and our experiment is the initial flow velocity of the plasma jets v_0 : 2000 km/s and ~ 200 km/s, respectively. From Liu et al., the growth rate Γ_{IFI} scales approximately linearly with the anisotropy of the jets A_i , given by

$$A_i = \frac{2v_0^2 + v_{ti,\parallel}^2}{v_{ti,\perp}^2} - 1 \sim 2 \frac{v_0^2}{v_{ti,\perp}^2},$$

where $v_{ti,\parallel}$ and $v_{ti,\perp}$ are the ion thermal velocity parallel and perpendicular to the flow velocity, respectively. Applying this flow velocity scaling leads to a growth time of ~ 10 ns for our experiment. As this is significantly longer than the counter-streaming time, we *do not* expect our plasma to be unstable to the IFI.

We believe that these findings are both interesting and strengthen our stated conclusions in the paper. As such we have included discussion of the possibility of IFI and Weibel instabilities driving EM turbulence to the results section, after the contrasting discussion of electrostatic turbulence on page 6. We consider “electromagnetic turbulence, which can be driven by plasma instabilities such as the ion Weibel instability and ion filamentation instability”, and show that “the counter-streaming phase is short-lived (3 ns) compared to the e-folding times of the IFI and Weibel instabilities (~ 10 ns)”, concluding that it is “unlikely that these instabilities contribute to field amplification and ion acceleration”.

Comment 4:

In Ren et al., Nature Communications 11, 5157 (2020), a similar experimental setup demonstrated enhanced stopping power of intense proton beams due to collective beam effects. The authors should also compare their findings with this work, emphasizing the role of heavy-ion beam flux as a key parameter.

Response:

We thank the reviewer for bringing this paper to our attention as it presents an interesting result that we had not previously considered. In Ren et al. (2020), PIC simulations and experimental measurement reveal the establishment of a longitudinal electric field induced by a beam-driven return current, enhancing the stopping-power beyond that expected with just binary collisions. However, this requires a high beam current ($3 \times 10^7 \text{ A/cm}^2$), compared to our chromium beam current of $\sim 10^{-2} \text{ A/cm}^2$. Similar simulations performed by Ren et al. with a lower current of $3 \times 10^2 \text{ A/cm}^2$ indicate an *absence* of this collective effect, with good agreement with the stopping power expected from binary collisions. As our current is safely below this threshold, we can treat our ion beam as a test beam and *do not* expect an enhanced stopping power due to collective effects. Nevertheless, we recognize that this is an important piece of information that is currently missing from the manuscript. In the description of the ion beam in the experimental setup section, we have elaborated that “each pulse contains approximately 9×10^5 ions, a sufficiently low number such that it can be treated as a test beam, with no collective effects expected”.

This comment has also motivated us to revisit the possibility of electrostatic turbulence driven by the ion beam itself. We consider the instability relevant to our experimental setup, the modified two-stream instability (MTSI), which can also excite electrostatic waves at the lower-hybrid frequency. The MTSI growth rate scales as

$$\gamma_{\text{MTSI}} \propto \sqrt{\frac{n_c m_c}{n_p m_i}},$$

where n_c and m_c are the beam density and beam ion mass, respectively; n_p and m_i are the plasma density and plasma ion mass, respectively. Given a low beam density of $\sim 10^7 \text{ cm}^{-3}$, we expect a MTSI growth rate on the order of $10^5 - 10^6 \text{ s}^{-1}$, much lower than the expected LHDI growth rate. For the MTSI to be competitive with the LHDI would require an ion beam density approximately 6 to 8 orders of magnitude higher, thus we can safely discount this possibility. We believe this analysis further strengthens our conclusions, and have included the following discussion in the results section: “An alternative avenue for the generation of electrostatic waves near the lower-hybrid frequency is via beam-driven instabilities, such as the modified two-stream instability (MTSI). To demonstrate that the ion beam acts as a test beam and does not excite plasma instabilities, we consider the MTSI growth rate: the ion beam density of $\sim 10^7 \text{ cm}^{-3}$ results in growth rates of the order $10^5 - 10^6 \text{ s}^{-1}$. This is much smaller than the growth rate for the lower-hybrid drift instability and therefore we discount the possibility of turbulence being driven by the ion beam.”